# Identification of Sugarcane Host Factors Interacting with the 6K2 Protein of the *Sugarcane Mosaic Virus*

**DOI:** 10.3390/ijms20163867

**Published:** 2019-08-08

**Authors:** Hai Zhang, Guangyuan Cheng, Zongtao Yang, Tong Wang, Jingsheng Xu

**Affiliations:** 1National Engineering Research Center for Sugarcane, Key Laboratory of Sugarcane Biology and Genetic Breeding, Ministry of Agriculture, Key Laboratory of Ministry of Education for Genetics, Breeding and Multiple Utilization of Crops, College of Crop Science, Fujian Agriculture and Forestry University, Fuzhou 350002, China; 2State Key Laboratory for Protection and Utilization of Subtropical Agro-bioresources, Guangxi University, Nanning 530004, China

**Keywords:** *sugarcane mosaic virus*, 6K2, yeast two-hybrid, interaction

## Abstract

The 6K2 protein of potyviruses plays a key role in the viral infection in plants. In the present study, the coding sequence of 6K2 was cloned from *Sugarcane mosaic virus* (SCMV) strain FZ1 into pBT3-STE to generate the plasmid pBT3-STE-6K2, which was used as bait to screen a cDNA library prepared from sugarcane plants infected with SCMV based on the DUALmembrane system. One hundred and fifty-seven positive colonies were screened and sequenced, and the corresponding full-length genes were cloned from sugarcane cultivar ROC22. Then, 24 genes with annotations were obtained, and the deduced proteins were classified into three groups, in which eight proteins were involved in the stress response, 12 proteins were involved in transport, and four proteins were involved in photosynthesis based on their biological functions. Of the 24 proteins, 20 proteins were verified to interact with SCMV-6K2 by yeast two-hybrid assays. The possible roles of these proteins in SCMV infection on sugarcane are analyzed and discussed. This is the first report on the interaction of SCMV-6K2 with host factors from sugarcane, and will improve knowledge on the mechanism of SCMV infection in sugarcane.

## 1. Introduction

Potyviruses, which account for 30% of known plant viruses, include many agriculturally important viruses, e.g., *Sugarcane mosaic virus* (SCMV), *Turnip mosaic virus*, *Tobacco etch virus*, and *Potato virus Y* [1,2]. The genomes of *Potyvirus* members consist of single-stranded, positive-sense RNAs of approximately 10,000 nucleotides that encode two polyproteins, which are self-cleaved into 11 mature proteins, which are P1, HC-Pro, P3, P3N-PIPO, 6K1, CI, 6K2, VPg, NIa-Pro, NIb, and CP [2,3,4,5,6,7]. To establish systemic infection on host plants, potyviruses have to employ the 11 functional proteins to interact with host factors and interplay with the cellular pathways [8].

The replication and intra/intercellular movement of the viral genome are key steps for the potyvirus to establish a systemic infection on host plants, in which 6K2 plays important roles. In the early stage of potyviral infection, 6K2 localizes to and remodels the endoplasmic reticulum (ER) membrane into convolutional structures, which mature and bud off into vesicles harboring the virus replication complex at ER exit sites in a coatomer protein I (COPI)-dependent and COPII-dependent manner [9,10,11]. These vesicles may fuse with chloroplasts for efficient replication [12]. The intracellular and intercellular movement of potyvirus is in the form of vesicles induced by 6K2 [13,14,15]. P3N-PIPO recruits CI to the plasmodesmata [5,10,14,16,17], while CI interacts with 6K2 and serves as a docking point for the intercellular movement of virus replication vesicles [14]. Besides involvement in potyviral replication and movement, 6K2 participates in the autophagy induced by potyvirus infection. 6K2 accompanied by VPg antagonizes the degradation of HC-Pro, thereby allowing potyviruses to participate in a trade-off with host antiviral autophagy [18,19].

Sugarcane (*Saccharum* spp. hybrid) is the most important sugar and energy crop worldwide. SCMV, *Sorghum mosaic virus* (SrMV), and *Sugarcane streak mosaic virus* are the main pathogens that cause severe mosaic disease in sugarcane and result in heavy yield loss in the sugarcane industry. However, the mechanism of mosaic pathogen infection on sugarcane remains in its infancy. Plant viruses are too simple in structure to establish systemic infection on host plants without interaction with host factors [8]. Considering that the potyviral 6K2 protein plays a key role in viral infection [9,10,11,12,13,14,15,17,18,19], we hypothesize that SCMV-6K2 might extensively interact with sugarcane host factors. In the present study, SCMV-6K2 was used as bait to screen a cDNA library prepared from SCMV-infected sugarcane plants by using yeast two-hybrid (Y2H) assays to identify the interacting proteins from sugarcane. This is the first report on the identification of the sugarcane host factors interacting with SCMV-6K2, and will benefit the pathogenesis of SCMV infection in sugarcane.

## 2. Results

### 2.1. Cloning and Subcellular Localization of SCMV-6K2

The coding sequence of SCMV-6K2 was cloned from the FZ1 strain. SCMV-6K2 is 53 amino acids in length, and contains a GXXXG motif (‘X’ being any amino acid) (Figure 1A) that is vital for potyvirus infection and is needed to produce replication vesicles [20]. TMHMM analysis showed that SCMV-6K2 is an intrinsic membrane protein consisting of a central transmembrane domain (Figure 1B). To determine the subcellular localization of SCMV-6K2, SCMV-6K2-CFP was coexpressed with the ER retention signal marker mCherry-HDEL in leaf epidermal cells of *N. benthamiana*. The results showed that the cyan fluorescent signal of SCMV-6K2-CFP merged with the red fluorescent signal of mCherry-HDEL on ER and the fluorescence of chlorophyll, indicating the ER or chloroplast localization of SCMV-6K2, respectively (Figure 1C), which is consistent with previous studies [12,14].

### 2.2. Construction and Evaluation of the pBT3-6K2 Bait Vector

Using *Sfi*I digestion, SCMV-6K2 was infused with the pBT3-STE plasmid to generate a pBT3-STE-6K2 recombinant plasmid. The yeast (*Saccharomyces cerevisiae*) strain NMY51 and synthetic dextrose (SD) medium SD/-Leu/-Trp (DDO) and SD/-Trp/-Leu/-His/-Ade (QDO) agar plates supplemented with 5-bromo-4-chloro-3-indolyl β-D-galactoside (X-Gal) were used in this study. Yeast cells co-transformed with pBT3-STE-6K2 and pOst1-NubI gave blue colonies, which indicated no toxicity of 6K2 to yeast cells (Figure 2). Yeast cells co-transformed with pBT3-STE-6K2 and pPR3-N produced colonies on DDO+X-Gal agar plates, but did not grow on QDO+X-Gal agar plates, which indicated no auto-activation of 6K2 (Figure 2). Therefore, the pBT3-STE-6K2 bait plasmid is suitable for cDNA library screening.

### 2.3. Screening of the Sugarcane cDNA Library and Gene Cloning

A total of 157 yeast colonies were collected from the QDO agar plates. Yeast plasmids were extracted and individually transformed into competent *E. coli* DH5α cells. Seventy cDNA fragments were obtained by colony PCR detection and sequentially sequenced. Based on the sequence information, special primer pairs (Appendix A) were designed to clone the corresponding full-length gene from sugarcane cv. ROC22. Homologous sequences of 29 genes were found in the National Center for Biotechnology Information or Phytozome v 12 by BLAST; however, five cDNA sequences were uncharacterized (data not shown). Therefore, the 24 genes and their putative coding proteins were used for further analysis (Table 1).

Based on the main functions, these 24 proteins were roughly classified into three categories: stress and defense response proteins, photosynthesis-related proteins, and transport-related proteins (Table 1). Eight proteins were involved in the stress and defense response. These proteins were ScH2A.2 (histone H2A.2), ScRNS4 (ribonuclease T2), ScULP5 (ubiquitin-like protein 5), ScSERINC3 (serine incorporator 3), ScVAMP727 (vesicle-associated membrane protein 727), ScTET18 (tetraspanin 18), ScPMP22 (peroxisomal membrane 22 kDa protein), and ScHSP82 (heat shock protein 82). Twelve proteins were involved in transport: aquaporins including ScPIP1; 2, ScPIP2; 7, and ScTIP1; 2, ScZIFL1 (zinc-induced facilitator-like 1), ScNCX1 (sodium/calcium exchanger 1), ScVHA-C (V-ATPase subunit C), ScSULTR3-3 (sulfate transporter 3-3), ScGONST4 (GDP-mannose transporter 4), ScPPT2 (phosphoenolpyruvate/phosphate translocator 2), ScTMEM208 (transmembrane protein 208), ScTPT (triose phosphate/phosphate translocator), and ScBGlu31 (beta-glucosidase 31). Four proteins were involved in photosynthesis: ScPsbS (photosystem II S subunit), ScPsbR (photosystem II R subunit), ScVTE3 (2-methyl-6-phytyl-1,4-hydroquinone methyltransferase), and ScCAB1 (chlorophyll a/b binding protein 1). The coding sequences of these genes were deposited into GenBank (Table 1).

#### 2.3.1. Stress and Defense Proteins

Eight proteins involved in stress and defense responses were screened from the cDNA library. Histone ScH2A.2, ScRNS4, and ScUPL5 were found to be involved in gene expression and regulation. H2A.2 is the variant of histone H2A, which is a highly conserved component of eukaryotic chromatin along with H2B, H3, and H4. There are 13 members of H2A in *Arabidopsis*. H2A.2 is mainly expressed in the veins of leaves and shows a low expression level, similar to that of H2A.1, in whole plants compared with other members [51]. Members of H2A variants have been implicated in the regulation of DNA repair, transcriptional activity, recombination, and response to water stress [21,52]. RNS4, an S-like RNase, belongs to the RNaseT2 family, in which many members are associated with defense responses [53]. The homologue OsRNS4 from rice is involved in responses to many biotic and abiotic stresses such as insect attacks, wounding, and infection from *Xanthomonas oryzae* or *Magnaporthe grisea* [22]. The homologue AtRNS3 from *Arabidopsis* plays a key role in the biogenesis of tRNA-derived RNA fragments, which are presumed to be involved in stress responses, cell proliferation regulation, and as a primer for viral reverse transcription [54]. Homologues of RNS4 from tobacco were upregulated under the challenge of *Phytophthora parasitica* [55] or the tobacco mosaic virus [23]. ULP5 is a component of the ubiquitin-proteasome system (UPS), in which the members play a critical regulatory role in most cellular processes [56,57,58,59]. Different from other ubiquitins, ULP5 contains an ubiquitin super-fold with the C-terminal sequence containing of a pair of tyrosines, but not a double glycine [24]. ULP5 was shown to mediate the susceptibility to strip rust in wheat at the seedling stage, and knockdown of the ULP5 increased the expression levels of the biotic stress-related genes *PR1* and *PR2* [25]. The yeast HUB1, which is a homologue of ULP5, is involved in the alternative splicing of pre-messenger RNAs by ubiquitination of the spliceosome [56]. 

VAMP727, SERINC3, and PMP22 are involved in biotic stress. VAMP727 is an R-SNARE (Soluble N-ethylmaleimide sensitive factor attachment protein receptor). SNAREs are the key regulators of the control trafficking of cargo proteins to their final destinations, and play a key role in plant development. VAMP727 is involved in vacuolar protein deposition, targeting of the plasma membrane of the brassinosteroid (BR) receptor BRI1 [60], and the response to root-knot nematode infection [26]. SERINC3 belongs to the serine incorporator (SERINC) family, which comprises five members, from SERINC1 to SERINC5, that are structurally characterized by having 11 transmembrane domains. SERINC proteins are involved in the biosynthesis of sphingolipids and phosphatidylserine by incorporating serine into membrane lipids [61]. SERINC3 and SERINC5 inhibit human immunodeficiency virus (HIV) infection as restriction factors [27]. However, the exact function of the SERINC family in plants is still unknown. PMP22 is an integral membrane protein of peroxisomes in all organs of the mature plant, and directly inserts into peroxisomes with the N-terminal and C-terminal parts facing the cytosol [28,62]. PMP22 is important for the biogenesis and function of the peroxisome [28,62]. Some tombusviruses deploy peroxisomes to generate multivesicular bodies for viral replication, which is a biological process that usually happens on the ER or chloroplast for potyviruses [63,64]. Peroxisomes can produce reactive oxygen species (ROS) to counteract viral pathogens [29,64,65].

TET18 belongs to a superfamily of small integral membrane proteins with four transmembrane domains. Most of the TETs localize to plasma membrane with several localizing to the ER [66]. There are 17 and 15 TETs in *Arabidopsis thaliana* and rice, respectively [66,67]. TETs interact with each other and other proteins to form TET-enriched microdomains, which are important in plants for development, reproduction, and stress responses [67,68]. Based on molecular mass, heat shock proteins can be classified into six families: Hsp100, Hsp90, Hsp70, Hsp60, Hsp40, and small heat shock proteins (sHSPs) [30]. HSPs are involved in the response to a wide variety of stresses including cold, drought, salt, UV light, wound, and biotic stresses [30]. HSP82 belongs to the Hsp90 family, in which the members promote the final maturation of a selected group of proteins including protein kinases, transcription factors, nuclear steroid receptors, and regulatory proteins [69].

#### 2.3.2. Transport-Related Proteins

Twelve transport-related proteins were screened from the cDNA library. Three aquaporins, ScPIP1; 2, ScPIP2; 7, and ScTIP1; 2, were identified. Aquaporins are highly conserved and present in all the living organisms [31]. Based on sequence similarity and subcellular localization, plant aquaporins are classified into five subfamilies: plasma membrane intrinsic proteins (PIPs), tonoplast intrinsic proteins (TIPs), nodulin26 (Nod26)-like intrinsic proteins (NIPs), small basic intrinsic proteins (SIPs) [32,33,70], and X-intrinsic proteins (XIPs) [71,72]. The management of PIP intracellular localization appears to be an important process by which plant cells modulate the plasma membrane water permeability [73]. PIPs reach their final destination via secretory pathway trafficking from the endoplasmic reticulum (ER) via the Golgi apparatus to the plasma membrane. PIP1; 2 and PIP2; 7 localize to the plasma membrane (PM), while TIP1; 2 localizes to the tonoplasts [32,33,70]. PIPs and TIPs are involved in the response to water stress [74,75], thereby affecting the assimilation of CO_2_ [76,77,78].

Three proteins involved in ion transport, ScZIFL1, ScNCX, and ScVHA-C, were screened. ZIFL1 is a transmembrane protein that belongs to the major facilitator superfamily [35]. In *Arabidopsis*, ZIFL1 localizes to the tonoplast, and is involved in Zn homeostasis [34], H^+^-coupled K^+^ transport, and polar auxin transport by modulating K^+^ and H^+^ fluxes [79]. The membrane-bound sodium–calcium exchanger (NCX) proteins regulate Ca^2+^ homeostasis in many cell types, thereby being involving in a wide range of physiological and pathological processes [80]. In rice, NCX1 contains one NCX domain and one EF-hand domain, and NCX1 was found to be highly expressed in all plant parts and at various developmental stages [36]. VHA-C (V-ATPase subunit C) is required for V-ATPase (vacuolar-type H+-ATPase) assembly and proton channel formation, and is directly responsible for the binding and transmembrane transport of protons in plant cells [81].

ScSULTR3-3, ScGONST4, ScPPT2, ScTPT, and ScBGlu31, which are involved in the transport of compounds or metabolic products, were screened. ScSULTR3-3 is a sulfur transporter (SULTR), which is a class of carrier proteins that is required for active sulfate transport [82,83]. There are five members of the SULTR3 family in *Arabidopsis*, and SULTR3-3 is localized to the chloroplast envelope [38]. The SULTR3 family are involved in sulfate transport into the chloroplast and influence sulfate assimilation, abscisic acid (ABA) biosynthesis, and sulfate-induced stomatal closure [38], thereby being involving in plant stress response [84]. ScGONST4, a GDP-mannose transporter, is a nucleotide sugar transporter that imports the nucleotide sugars synthesized in either the cytoplasm or the nucleus into the Golgi and ER lumen across the organellular membranes for glycosylation [85]. There are five members of the GDP-mannose transporter family in *Arabidopsis*, which are all localized to the Golgi apparatus [39]. In *Saccharomyces cerevisiae*, the VRG4 protein, the homologue of GONST4, transports GDP-mannose into the Golgi for mannosylation. Mutant *Candida albicans* VRG4 strains have defective hyphal formation. Since the *VRG4* gene is essential for yeast viability but does not have a mammalian homologue, it is a particularly attractive target for the development of antifungal therapies [86]. ScPPT2 is localized to the inner envelope membrane of both non-green plastids and chloroplasts, and it reverse exchanges phosphoenolpyruvate (PEP) with inorganic phosphoric acid [40]. PPT (phosphoenolpyruvate/phosphate translocator) belongs to the family of phosphate translocators that are localized to the inner envelope membrane of both non-green plastids and chloroplasts and reverse exchanges phosphoenolpyruvate (PEP) with inorganic phosphoric acid [40]. Mutants deficient in PPT2 slightly retard the growth of *Arabidopsis* plants [87]. TPT (triose phosphate/phosphate translocator) is localized to the chloroplast inner envelope and plays key roles in photosynthesis by catalyzing the strict 1:1 exchange of triose-phosphate, 3-phosphoglycerate, and inorganic phosphate across the chloroplast envelope [41,42]. The mutation of *tpt* significantly reduces the maximum rate of O_2_ evolution in CO_2_-saturated conditions in *Arabidopsis* plants [88]. BGlu31 is localized to the vacuole, and is involved in glucose metabolism to supply acyl and glucosyl for glycosylation in rice [43].

TMEM208 is an ER-located protein that is involved in both autophagy and ER stress [44]. TMEM208 is highly conserved among eukaryotes and represents the third pathway, i.e., the SND (SRP-independent targeting) pathway, for transporting proteins into the ER besides the SRP (Signal Recognition Particle) and GET (Guided Entry of Tail-anchored proteins) pathways [89].

#### 2.3.3. Photosynthesis-Related Proteins

Four proteins that are directly involved in photosynthesis were screened. The PsbS monomers form dimers at a high lumen pH, whereas they stay as monomers at a low pH, thereby allowing the monomeric PsbS interacting with LHCII (light-harvesting chlorophyll a/b-binding protein complex II) [45,46] to activate the non-photochemical quenching (NPQ), thereby protecting the photosynthetic organisms against excess light by dissipating the excess absorbed energy into heat [90,91]. PsbR, along with PsbO, PsbP, and PsbQ, are the four extrinsic subunits of the oxygen-evolving complex (OEC) protein of the PSII in higher plants [47,48]. PsbR is involved in binding PsbP [92], and is essential for the optimal oxygen-evolving activity of PSII [47,93,94]. PsbR is required for the binding of LHCSR3 to PSII-LHCII supercomplexes, the efficiency of NPQ in *Chlamydomonas reinhardtii* [95], and the tolerance to cold stress in rice [96]. VTE3 is involved in a key methylation step in both vitamin E and plastoquinone synthesis [97]. VTE3 is localized at the chloroplast envelope membrane, and plays an important role in chloroplast development [49]. The knockout of *VTE3* causes mutants with a pale green phenotype, abnormal chloroplasts, and non-survival beyond the seedling stage [49,97].

The light-harvesting chlorophyll a/b binding protein 1 (LHCB1) binds the pigments chlorophyll a, b, and carotenoids to form the LHCIIb complex, which targets the thylakoid membrane, thereby playing a key role in the photosynthesis [50].

### 2.4. Verification of the Interaction between the Screened Proteins and SCMV-6K2

To verify whether SCMV-6K2 interacts with the 24 proteins identified based on the library screen, the Y2H technique was applied. The cDNAs of the 24 proteins were individually constructed into the prey vector pPR3-N to generate the pPR3-N fusion plasmids. Then, the pPR3-N fusion plasmid was co-transformed with the bait vector pBT-STE3-6K2 into yeast NMY51 cells and cultured on DDO and QDO culture media supplemented with X-Gal. The results showed that the yeast cell co-transformations of pBT-STE3-6K2 with pPR3-N-ScBGLU31, pPR3-N-ScHSP82, pPR3-N-ScCAB1, and pPR3-N-ScTPT could not produce blue colonies on the DDO and QDO culture media. Yeast cells co-transformed in pairs with other prey vectors and the bait vector pBT-STE3-6K2, and the positive control gave blue colonies on DDO and QDO media supplemented with X-Gal (Figure 3).

## 3. Discussion

The plant virus genome is too simple to establish systemic infections on host plants without interacting or interplaying with host factors and cellular processes [8]. The higher eukaryotic plants employ a complicated membrane system to compartmentalize the biological process in different organelles. When infected by compatible positive strand RNA viruses, the membranes of organelles, including the ER, chloroplasts, vacuoles, peroxisomes, mitochondria, Golgi, and endosomes, are drastically rearranged to form viral replication complexes [98]. In the present study, 20 proteins were identified from sugarcane to interact with SCMV-6K2 by Y2H assay (Table 1) and were classified into three categories based on their functions, i.e., eight proteins in the stress and response group, 12 proteins in the transport group, and four proteins in the photosynthesis group. Since 6K2 is an integral membrane protein and the screening of cDNA library was based on the membrane system, most of the identified proteins are localized to PM or the membranes of the chloroplasts, vacuoles, or peroxisomes (Table 1). 

Although SCMV is compatible with the sugarcane cultivar ROC22, SCMV infection causes a defense response in sugarcane. Seven proteins interacting with SCMV-6K2 by Y2H assays are involved in the stress and defense response (Table 1): RNS4, ULP5, VAMP727, SERINC3, PMP22, and TET18. The stress responses of the sugarcane plant to SCMV infection include the regulation of gene expression, signal transduction, and the ROS reaction. HTA2, RNS4, and UPL5 are involved in DNA repair, transcription, recombination [21,52], microRNA biogenesis [54], and alternative splicing of pre-mRNA [56], respectively. Thus, the interaction of SCMV-6K2 with these three proteins might interfere with gene expression and sugarcane plant regulation. VAMP727 and SERINC3 are involved in signal transduction. VAMP727 interacts with SYP22 in response to root-knot nematode infection via the control of abundances of BRI1 on the PM [26,60], while SERINC3 is involved in the biosynthesis of sphingolipids and phosphatidylserine [99], which are the main components of PM [61], with the former playing a key role in the formation of the membrane raft [100]. The membrane raft is the main signal transduction platform on PM, and is extensively involved in cell processes including response to viral infection [101]. Therefore, the interaction of SCMV-6K2 with VAMP727 or SERINC3 might interfere with the BR signal cascade or the membrane raft, respectively, to respond to SCMV in sugarcane plants. TET18 might be a component of membrane raft, as TETs interact with each other and other proteins to form TET-enriched microdomains [67,68]; thus, we presume that TET18 might be involved in signal transduction or membrane contact between the VRC and the ER or PM. PMP22 localizes to peroxisomes and is important for the biogenesis and function of the peroxisomes [28,62]. Some tombusviruses deploy peroxisomes to generate multivesicular bodies for viral replication, which is a biological process that usually happens on the ER or chloroplast for potyviruses [63,64]. Peroxisomes can produce ROS to counteract viral pathogens [29,64,65]. The interaction of SCMV-6K2 with PMP22 might interfere with the biogenesis, or at least with the function of peroxisomes, to suppress the production of ROS, thereby facilitating viral infection. 

Ten transport-related proteins—aquaporins including PIP1; 2, PIP2; 7 and TIP1; 2, ZIFL1, NCX1, GONST4, SULTR3-3, PPT2, VHA-C, and TMEM208—were found to interact with SCMV-6K2 by Y2H assays. These proteins are involved in the transport of water, ions, metabolites, and proteins, which are important for maintaining the normal biological process. Three aquaporins, PIP1; 2, PIP2; 7, and TIP1; 2, were identified. PIPs and TIPs are involved in the response to water stress [73,74], thereby affecting the assimilation of CO_2_ [76,77,78]. The interaction between SCMV-6K2 with these aquaporins may limit the water supplement for CO_2_ assimilation during photosynthesis, thereby inhibiting the growth of infected plants. ZIFL1, NCX1, and VHA-C are involved in the homeostasis of H^+^, Zn^2+^, K^+^, Na^+^, and Ca^2+^ [34,35,79], while ZIFL1 is also involved in the polar transport of Auxin [79]. VHA-C is a key subunit of V-ATPase, which plays an important role in the acidification of subcellular organelles, pH and ion homeostasis, and endocytic and secretory trafficking [81]. PutVHA-C from *Puccinellia tenuiflora* is distributed throughout the secretory pathway, as the endosomes containing PutVHA-C can fuse with each other and simultaneously transport and fuse with the plasma membrane, tonoplast, and cell plate [81]. The overexpression of PutVHA-C enhances V-ATPase activity and promotes plant growth by influencing V-ATPase-dependent endosomal trafficking in transgenic *Arabidopsis* [37]. Therefore, the interaction of SCMV-6K2 with VHA-C might interfere with the transport of membranes for cell growth, membrane proteins, extracellular proteins, and components of the cell wall, thereby exerting an influence on the plant cell growth. The interaction of SCMV-6K2 with ZIFL1 or NCX1 might interfere with the homeostasis of Zn^2+^, K^+^, Na^+^, and Ca^2+^, or the transport of Auxin, thereby impacting the normal biological process, such as through the regulation of vial cell-to-cell movement, as Ca^2+^ has been extensively reported to be involved in virus infections in plants by regulating callus accumulation or degradation in the plasmodesmata [5]. The interaction of SCMV-6K2 with GONST4, SULTR3-3, or PPT2 might interfere with the transport of GDP-mannose, sulfate, and phosphoenolpyruvate, thereby interfering with mannosylation in the Golgi or photosynthesis, ABA biosynthesis [38], and the shikimate pathway [40,102] in chloroplasts. In addition [74], SULTR3; 3 is involved in sulfate-induced stomatal closure [38], thereby being involved in plant stress and the plant stress response [84]. PPT2 might mediate the fusion of VRC with the envelope of chloroplast, producing efficient multiplication [15], as mutants deficient in PPT2 slightly retard the growth of *Arabidopsis* plants, leaving photosynthesis, leaf constituents, and transport unaffected [87]. TMEM208 is localized to the ER, and is involved in the transport of proteins with the transmembrane domain in the central position [89]. Interestingly, the transmembrane domain is in the central position of 6K2 from potyviruses [20]. The ER is very important in the viral infection of plants. Usually, viral infection causes ER stress and an unfolded protein response [103]. SCMV-6K2 might interfere with protein transport into the ER, autophagy, or ER stress via interaction with TMEM208 [44]. However, the exact function of TMEM208 has not been identified in plants. 

Chloroplasts are important organelles for photosynthesis. The potyvirus 6K2-mediated VRC vesicle fuses with the outer envelope of the chloroplast for efficient multiplication [15]. Three proteins are directly involved in photosynthesis: PsbS, PsbR, and VTE3. The interaction of SCMV-6K2 with PsbS and PsbR may impair the interaction of monomeric PsbS with LHCII [45,46], or with the binding of LHCSR3 to PSII-LHCII supercomplexes [95], thereby inhibiting the efficiency of NPQ [90,95,104]. The interaction of SCMV-6K2 with PsbR might interfere with the oxygen-evolving activity of PSII, as PsbR is required for the assembly of the oxygen-evolving complex [47,48,93,94]. The interaction of SCMV-6K2 with VTE3 might impair the synthesis of vitamin E and plastoquinone, which are required functional chloroplasts [49,97]. VTE3 knockout causes mutants with a pale green phenotype, abnormal chloroplasts, and non-survival beyond the seedling stage [49,97]. Considering that these three proteins are important for functional chloroplasts, their interaction with SCMV-6K2 might contribute to mosaic symptoms, thereby attracting aphids for SCMV dissemination.

The results of cDNA library screening and the interaction between SCMV-6K2 and the identified proteins verified our hypothesis that SCMV-6K2 extensively interacts with sugarcane host factors. The identification of 20 proteins that interact with SCMV-6K2 enhances the understanding of the macular mechanism of SCMV infection on sugarcane and provide potential molecular targets for genetic improvement of sugarcane germplasm or cultivar. Since the genomic structure of SCMV is similar to that of SrMV or SCSMV [5,103,105], this study might increase the knowledge on the mechanism of SrMV or SCSMV infection on sugarcane. However, it should be noted that the evidence for interaction is based on Y2H assays, which indicate the interaction in vitro. To confirm the interaction of SCMV-6K2 with these 20 proteins in vivo, bimolecular fluorescence complementation or coimmunoprecipitation should be performed. In addition, the library might not cover all the genes that interact with SCMV-6K2. The interaction of SCMV-6K2 with the sugarcane host factors might be dynamic and affected by the developmental stage of sugarcane plants or the infection process of SCMV infection in sugarcane plants. The subsequent work will focus on the verification of the SCMV-6K2 interacting proteins in vivo and their functions in SCMV infection in sugarcane. 

## 4. Materials and Methods

### 4.1. Materials and Plant Culture

A cDNA library prepared from the leaves of sugarcane cultivar ROC22 infected by SCMV and SCMV strain FZ1 (KR108212) [105] were provided by the Fujian Key Laboratory of Sugarcane Biology and Genetic Breeding, Ministry of Agriculture, Fujian Agriculture and Forestry University (Fujian, China). SCMV-FZ1 was propagated on sugarcane cv. ROC22, which was planted in a greenhouse under controlled temperature (28 °C) and relative humidity (60%) conditions with a 14–10 h light–dark cycle. Sugarcane plant leaves with typical mosaic symptoms (Appendix A) were put into liquid nitrogen immediately after sampling and transferred to the −80 °C refrigerator for RNA isolation. *N. benthamiana* plants were grown in soil at 22 °C and 60% relative humidity under long days (16 h light/8 h dark). Two-week-old *N. benthamiana* plants were used for agroinfiltration experiments. After agroinfiltration, the plants were maintained under the same growth conditions.

### 4.2. RNA Isolation and Gene Cloning

Total RNA was extracted from the leaves of SCMV-FZ1-infected sugarcane cultivar ROC22 using TRIzol reagent (Invitrogen, New York, NY, USA) according to the manufacturer’s instructions. The RNA concentration was determined using Nanodrop (Thermo Scientific, Shanghai, China), and the first-strand cDNA was synthesized by using the Prime Script RT Reagent Kit (TaKaRa, Dalian,China). Specific primer pairs were designed based on the sequences of target genes using cDNA as the template. 

### 4.3. Bioinformatic Analysis

The gene sequence or corresponding deduced amino acid sequence was used as a query to search the National Center for Biotechnology Information database (https://www.ncbi.nlm.nih.gov/) or Phytozome v.12 (https://phytozome.jgi.doe.gov/pz/portal.html#). TMHMM Server v. 2.0 (http://www.cbs.dtu.dk/services/TMHMM/) was used to predict the transmembrane domain of the target protein [106].

### 4.4. Plasmid Construction

To construct the plasmids for subcellular localization, the coding sequence of SCMV-6K2 was amplified with a special primer pair (Appendix A) and infused into the plasmid pEarleyGate102 to generate SCMV-6K2-CFP.

To construct the SCMV-6K2 bait plasmid, the coding sequence of SCMV-6K2 was amplified using primers 6K2-F and 6K2-R (Appendix A). The PCR product was recovered and subcloned into the pPOTO-Blunt Simple vector (Aidlab, Beijing, China), which was used as a template to amplify the coding sequence of 6K2 with the primer pair pBT3-STE-6K2-F and pBT3-STE-6K2-R (Appendix A). The amplified fragments were digested with the Sfi I enzyme (Thermo Scientific, Shanghai, China) and fused into the pBT3-STE plasmid to generate the pBT3-STE-6K2 plasmid. The same procedure was followed to construct the prey plasmid with the target gene infused into the plasmid pPR3-N. All the recombinant plasmids in this study were confirmed by DNA sequencing.

### 4.5. Transient Protein Expression and Confocal Microscopy

Agrobacterium tumefaciens-mediated transient protein expression assays of *N. benthamiana* leaves were performed as described previously [5]. SCMV-6K2-CFP and mCherry-HDEL were co-agroinfiltrated into the leaves of *N. benthamiana* using needleless syringes. The agroinfiltrated plants were maintained under normal growth conditions for 48 to 72 h. For confocal microscopy analysis, plant samples expressing recombinant proteins were imaged using a Leica SP8 confocal microscope (Leica Microsystems, Beijing, China) with an Argon laser. CFP was excited with 442-nm laser lines, and the emitted light was captured at 450–500 nm. mCherry was excited with 552-nm laser lines, and the emitted light was captured at 590–630 nm. The fluorescence of chlorophyll was excited with 552-nm laser lines, and the emitted light was captured at 650–680 nm. Images were captured digitally and processed with LSM software.

### 4.6. Evaluation of the SCMV-6K2 Bait Plasmid

To evaluate whether the plasmid pBT3-STE-6K2 is suitable for cDNA library screening, yeast cells were co-transformed with the pBT3-STE-6K2 and pOst1-NubI pair or the pBT3-STE-6K2 and pPR3-N pair, respectively. Yeast cells co-transformed with pTSU2-APP and pNubG-Fe65 were used as positive controls; pTSU2-APP and pPR3-N were used as negative controls. The co-transformed yeast cells were cultured on DDO agar plates at 30 °C for 3–5 days. Then, the colonies were suspended in DDO liquid medium to an OD_600_ of 0.6. A 10× dilution series of 10-μL aliquots of co-transformed NMY51 cells were spotted onto DDO and QDO agar plates supplemented with X-Gal, and they were incubated at 30 °C for 3–5 days. If yeast cells co-transformed with pBT3-STE-6K2 and pOst1-NubI gave blue colonies on the DDO and QDO agar plates, this indicated that the expression 6K2 was not toxic to yeast cells. If yeast cells co-transformed with pBT3-STE-6K2 and pPR3-N gave colonies on the DDO but did not grow on QDO agar plates, this indicated that there was no auto-activation of 6K2. 

### 4.7. Screening of the cDNA Library and Positive Colony Sequencing

The yeast NMY51 strain was used to identify interactions between expressed proteins. The plasmid pBT3-STE-6K2 was used as bait to screen the cDNA library based on the DUALmembrane system (Clontech, Mountain View, CA, USA), as described by Song et al. [107]. The colonies that grew well in QDO liquid medium were harvested. Plasmids were extracted using the TIANprep Yeast Plasmid DNA Kit and transformed into competent E. coli DH5α cells. Then, the E. coli cells were grown on LB medium with 50 μg·mL^-1^ of ampicillin to identify the transcription-activating domain (AD)/library plasmids. Six colonies were randomly selected from each plate for PCR detection using the primer pair pPR3-N-F and pPR3-N-R (Appendix A). The amplified fragments were recovered and sequenced for identification.

### 4.8. Verification of Protein Interaction by Y2H Assays

The DUALmembrane system was used in accordance with the manufacturer’s protocols. The prey vector pPR3-N, infused with the target gene to be tested, and the bait vector pBT3-STE-6K2 were co-transformed pairwise into the yeast strain NMY51. Yeast cells were spread on DDO agar plates and incubated at 30 °C for 3–5 days after transformation. Colonies grown on DDO plates were suspended in DDO liquid medium to an OD_600_ of 0.6. A 10 × dilution series of 10-μL aliquots of co-transformed NMY51 were spotted onto DDO and QDO agar plates supplemented with X-Gal to test the expression of the LacZ marker. Plates were incubated at 30 °C for 3–5 days. pTSU2-APP and pNubG-Fe65 interact in the Y2H assay and were used as positive controls. pPR3-N and pTSU2-APP do not form complexes, and were used as negative controls. All the Y2H assays were performed in triplicates.

## Figures and Tables

**Figure 1 ijms-20-03867-f001:**
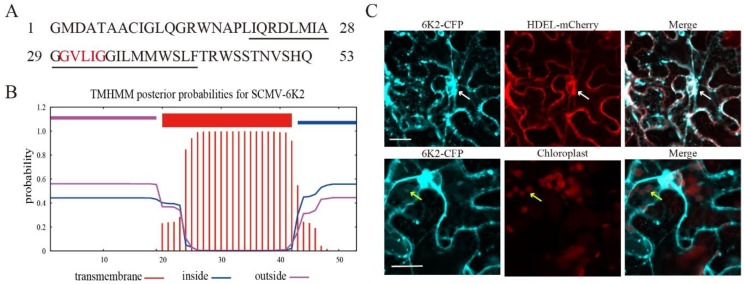
Subcellular localization of *Sugarcane mosaic virus* (SCMV)-6K2. (**A**) The schematic diagram of the amino acids of the SCMV-6K2 protein. GXXXG motif (‘X’ being any amino acid) was highlighted by the red color. The predicted transmembrane domain (TMD) was marked by an underline. (**B**) Prediction of SCMV-6K2 TMD by TMHMM. The horizontal axis indicates the amino acid position. (**C**) Subcellular localization of 6K2-CFP in the leaves epidermal cells of *N. benthamiana* by 48-h post agroinfiltration. White arrows point to endoplasmic reticulum (ER), yellow arrows point to chloroplast. Scale bars, 25 μm.

**Figure 2 ijms-20-03867-f002:**
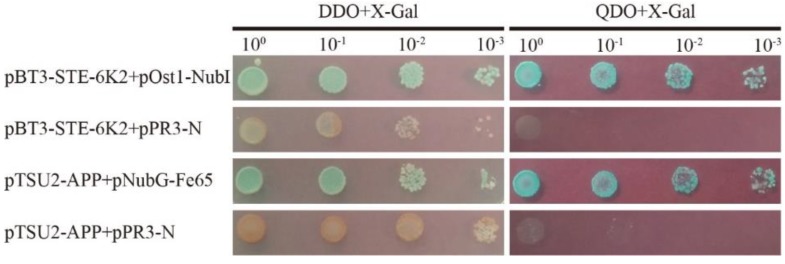
Evaluation of the pBT3-STE-6K2 bait vector. Plasmid combinations of pBT3-STE-6K2 and pOst1-NubI, pBT3-STE-6K2, and pPR3-N were co-transformed into yeast NMY51 cells in a 10× dilution series of 10-μL aliquots, which were then cultured on DDO+5-bromo-4-chloro-3-indolyl β-D-galactoside (X-Gal) or QDO+X-Gal agar plates to evaluate the toxicity or auto-activation of 6K2, respectively. Yeast cells co-transformed with pTSU2-APP and pNubG-Fe65 were used as positive controls, pTSU2-APP and pPR3-N were used as negative controls. DDO+X-Gal: SD/-Trp/-Leu, supplemented with X-Gal,; QDO+X-Gal: SD/-Trp/-Leu/-His/-Ade, supplemented with X-Gal.

**Figure 3 ijms-20-03867-f003:**
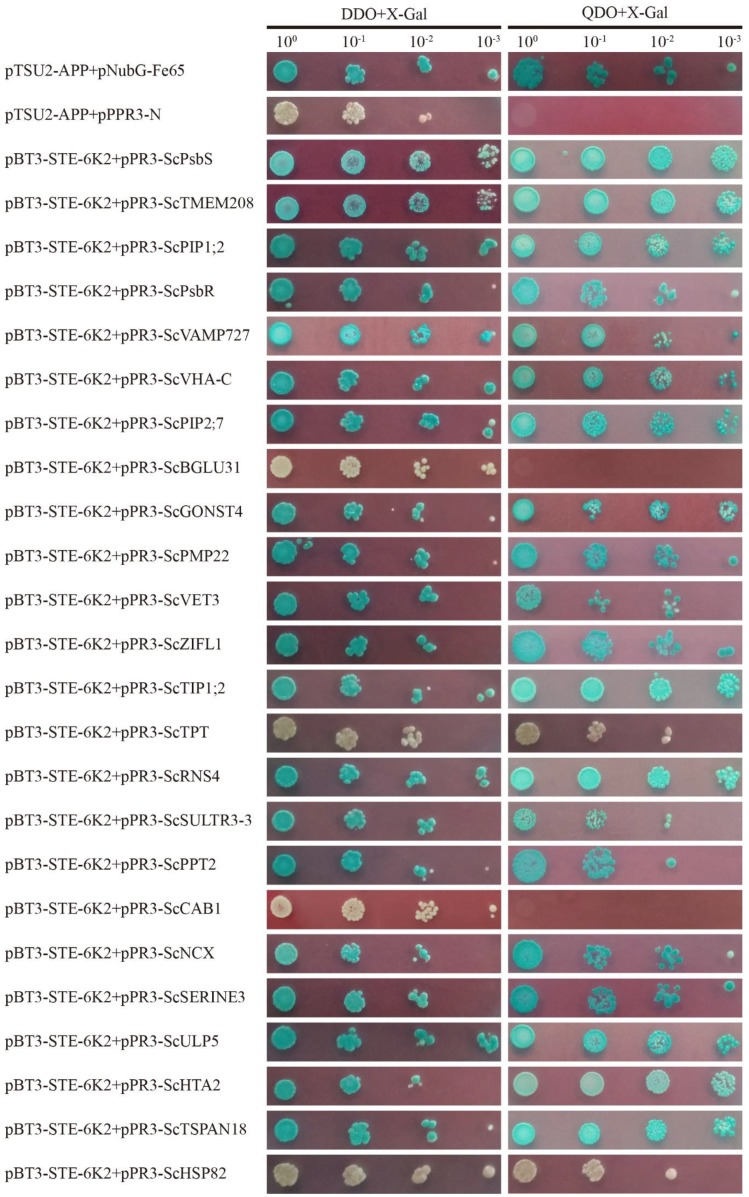
Verification of protein interaction by yeast two hybrid assays. The coding sequences of 24 proteins were individually infused into the prey vector pPR3 and co-transformed with the bait vector pBT3-STE-6K2 into the yeast NMY51 cells in a 10× dilution series of 10-μL aliquots, which were then plated on non-selective medium (DDO+X-Gal) or a high-stringency selective medium (QDO+X-Gal). Yeast cells co-transformed with pTSU2-APP and pNubG-Fe65 were used as positive controls, pTSU2-APP and pPR3-N were used as negative controls.

**Table 1 ijms-20-03867-t001:** Classification of proteins interacting with SCMV-6K2.

Protein Name (Coding Protein)	Accession Number	Specie and Accession Number of Homologue	Functional Description	Homology (%)	Clone Number	Subcellular Location	Reference
**Stress and Defense Response**							
ScHTA2 (Histone H2A.2)	MN16790	*Sorghum bicolor* XM_002460779.2	Involved in DNA repair, transcriptional activity, recombination	96.81	1	N	[21]
ScRNS4 (Ribonuclease T2)	MN167902	*Sorghum bicolor* XM_002462696.2	Biogenesis of tRNAs derived RNA fragments and viral reverse transcription	94.36	1	N	[22]
ScULP5 (Ubiquitin-like protein 5)	MN167908	*Setaria italica *XM_004977457.2	Involved in mRNA splicing and cellular protein modification	99.10	1	Cyto, N	[23,24]
ScVAMP727 (Vesicle-associated membrane protein 727)	MN167893	*Sorghum bicolor* XM_021465361.1	Involved in the vacuolar protein deposition and brassinosteroids receptor BRI1 PM targeting	96.83	2	V	[25]
ScSERINC3 (Serine incorporator 3)	MN167907	*Sorghum bicolor* XM_002454643.2	Synthesis of phosphatidylserine and sphingolipid	98.15	1	PM, ER	[26]
ScPMP22 (Peroxisomal membrane 22 kDa protein)	ScPMP22	*Sorghum bicolor* XM_002466831.2	Import of peroxisomal matrix proteins and the transport of metabolites across the membrane	96.54	3	P	[27,28]
ScTSPAN18 (Tetraspanin 18)	MN167910	*Sorghum bicolor* XM_002437610.2	Involved in the development, reproduction, and stress responses in plants	96.16	1	PM	[29]
ScHSP82 (Heat shock protein 82)	MN167911	*Sorghum bicolor* XM_002447368.2	Involved in cellular response to heat, protein folding, and protein stabilization	95.22	1	Cyto	[30]
**Transport-Related Proteins**							
ScPIP1; 2 (Aquaporin PIP1-2)	MN167891	*Sorghum bicolor * XM_002454463.2	Maintenance of cellular water homeostasis	99.08	1	PM	[31,32,33]
ScPIP2; 7 (Aquaporin PIP2-7)	MN167895	*Sorghum bicolor* XM_021453930.1	Maintenance of cellular water homeostasis	93.11	4	PM	[31,32,33]
ScTIP1; 2 (Aquaporin TIP1-2)	MN167901	*Sorghum bicolor* XM_002459138.2	Maintenance of cellular water homeostasis	94.79	1	V	[31,32,33]
ScZIFL1 (Zinc-induced facilitator-like 1)	MN167900	*Sorghum bicolor* XM_002457622.2	Maintenance of Zn homeostasis. Transport of [34] H+-coupled K+ and the polar transport of auxin	96.08	1	V, PM	[35]
ScNCX (Sodium/calcium exchanger)	MN167906	*Sorghum bicolor* XM_021457777.1	Maintenance of Ca^2+^ homeostasis	95.14	1	PM	[36]
ScVHA-C (V-type proton ATPase C subunit)	MN167894	*Sorghum bicolor* XM_002441847.2	Required for the assembly of V-ATPase and proton channel formation and transmembrane transport of protons	96.39	1	V	[37]
ScSULTR3-3 (Sulfate transporter 3-3)	MN167903	*Sorghum bicolor* XM_002448617.2	Transport of sulfate into chloroplast	96.48	1	Ch	[38]
ScGONST4 (GDP-mannose transporter 4)	MN167897	*Sorghum bicolor* XM_021454048.1	Transport of GDP-mannose into Golgi for protein glycosylation	96.67	1	V	[39]
ScPPT2 (Phosphoenolpyruvate/phosphate translocator 2)	MN167904	*Sorghum bicolor* XM_002454771.2	Transmembrane transport of phosphoglycerate	94.73	1	Ch	[40]
ScTPT (Triose phosphate/phosphate translocator)	MN167912	*Sorghum bicolor* XM_002454822.2	Transport of triose phosphates derived from the Calvin cycle in the stroma into the cytosol for sucrose synthesis and other biosynthetic processes	97.54	1	Ch	[41,42]
ScBGLU31 (beta-glucosidase 31)	MN167896	*Sorghum bicolor* XM_002438540.2	Involved in the glucose metabolism to supply acyl and glucosyl for glycosylation	90.61	1	V	[43]
ScTMEM208 (Transmembrane protein 208)	MN167890	*Sorghum bicolor* XM_002452735.2	Transport of protein into ER	97.52	1	ER	[44]
**Photosynthesis**							
ScPsbS (Photosystem II S subunit)	MN167889	*Sorghum bicolor* XM_002456659.2	Involved in nonphotochemical quenching	96.41	3	Ch	[45,46]
ScPsbR (Photosystem II R subunit)	MN167892	*Sorghum bicolor* XM_002443957.2	Involved in the assembly of PSII, particularly that of the oxygen-evolving complex subunit PsbP	94.66	1	Ch	[47,48]
ScVTE3 (2-methyl-6-phytyl-1,4-hydroquinone methyltransferase 2)	MN167899	*Sorghum bicolor* XM_002443518.2	Involved in the synthesis of vitamin E and plastoquinone	94.38	3	Ch	[49]
ScCAB1 (Chlorophyll a/b binding protein 1)	MN167905	*Sorghum bicolor* XM_002455856.2	Involved in the formation of the LHCIIb complex	97.11	1	Ch	[50]

Note: Cyto, cytoplasm; Ch, chloroplast; PM, plasma membrane; P, peroxisome; N, nucleus; V, vacuole.

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
