# Peer review of "Identification of Sugarcane Host Factors Interacting with the 6K2 Protein of the Sugarcane Mosaic Virus"

_ijms, 2019, doi:10.3390/ijms20163867_

Round 1

Reviewer 1 Report

The in vitro results may not extrapolate to the situation in vivo. The ‘interacting’ host proteins might turn out not to interact with the virus 6K2 protein in vivo. I suggest that the authors should leave other possibilities open.

Specifically, the manuscript could be improved by paying attention to the following:

The title might not be appropriate since the ‘interaction’ has not been verified in vivo.

The ‘interacting’ host factors include the proteins residing in thylakoid membrane and the proteins in nucleus. Considering that SCMV replicates in cytosol and makes replication complexes on ER, it is difficult to imagine how the viral protein interacts with those proteins deep inside the chloroplast or in nucleus. In the Discussion, the authors describe as if the host proteins identified in this study are actually ‘interacting’ with the 6K2 in vivo. I think it is inappropriate. Whether the host factors interact with the 6K2 in plant cells should be verified by an in vivo interaction study.

Please check any grammatical errors. I noticed two such errors in Figure. 1 legend.

Author Response

We would like to thank the Reviewer for his or her extensive comments and suggestions that significantly improved this manuscript. In the revised version of the manuscript, we have made every attempt to address all comments and implement all suggestions recommended by the Reviewer.

Point 1: The in vitro results may not extrapolate to the situation in vivo. The ‘interacting’ host proteins might turn out not to interact with the virus 6K2 protein in vivo. I suggest that the authors should leave other possibilities open.

Response: Thanks for your advice. We agree with you as these interacting host proteins were identified by Y2H, a technique which has defects such as false positive. We had mentioned this at the end of the Discussion section. However, we should not as confirmative as in the previous submitted version. We revised the manuscript and emphasized the possibility of false positive. Please check Line 268, 291-293, 344-350.

Point 2: The title might not be appropriate since the ‘interaction’ has not been verified in vivo.

Response: We agree with you. We had considered to title the manuscript as 'Identification of Sugarcane host factors interacting with the 6K2 protein of the Sugarcane mosaic virus by yeast two hybrid' or 'Screening of Sugarcane host factors interacting with the 6K2 protein of the Sugarcane mosaic virus by yeast two hybrid'. However, these two titles are not as concise as the one we used in the submitted manuscript. We noticed that Song et al. published a paper titled as 'Identification for Soybean host factors interacting with the P3N-PIPO protein of the Soybean mosaic virus' (Acta Physiol Plant. 2016, 38:131. DOI 10.1007/s11738-016-2126-6). In this paper, the authors did not clone the corresponding genes or verify the interaction. They just showed the positive yeast colonies growing on medium. In our manuscript, we cloned the interacting genes and verified the interaction with SCMV-6K2 one by one by Y2H assays. We revised the main text to make it more clear that only Y2H evidence was provided and further experiments such as BiFC or Co-IP should be performed to confirm the interaction. Please check Line 268, 291-293, 344-350.

Point 3: The ‘interacting’ host factors include the proteins residing in thylakoid membrane and the proteins in nucleus. Considering that SCMV replicates in cytosol and makes replication complexes on ER, it is difficult to imagine how the viral protein interacts with those proteins deep inside the chloroplast or in nucleus.

Response: We agree with you. It is surprising to see that SCMV-6K2 interacts with the proteins localized to the inner of chloroplast or nucleus resulting from Y2H assays. Plant virus are too simple in structure to establish systemic infection on host plants without interaction with host factors of employing the cellular processes. Song et al. screened the soybean cDNA library with P3N-PIPO of Soybean mosaic virus (SMV) as bait by Y2H and obtained several proteins which localize to the inner of chloroplast or nucleus (Acta Physiol Plant. 2016, 38:131. DOI 10.1007/s11738-016-2126-6). For example, PsbR and thioredoxin-like protein HCF164 are localized to the thylakoid membrane. The localization of HCF164 was experimentally verified by Motohashi et al. (HCF164 Receives Reducing Equivalents from Stromal Thioredoxin across the Thylakoid Membrane and Mediates Reduction of Target Proteins in the Thylakoid Lumen. J Biol Chem, 2006, 281(46):35039-35047). The nucleus localization of ATHB6, homologue of homeodomain-leucine zipper protein 56 isoform X1 (ATHB16) which interacts with SMV-P3N-PIPO, was also experimentally verified by Lechner et al. (MATH/BTB CRL3 receptors target the homeodomain-leucine zipper ATHB6 to modulate abscisic acid signaling. Dev Cell, 2011, 21(6):1116-1128). We doubly checked the chloroplast or nucleus inner localization of the proteins mentioned above by searching the website http://suba.live/index.html. In the present study, we didn’t discuss how SCMV-6K2 enter the inner of chloroplast or nucleus. We believe that illumination of the underlying mechanism of viral protein interacting with host factors in the inner organelles is of biological and scientific significances.

Point 4: In the Discussion, the authors describe as if the host proteins identified in this study are actually ‘interacting’ with the 6K2 in vivo. I think it is inappropriate. Whether the host factors interact with the 6K2 in plant cells should be verified by an in vivo interaction study.

Response: Many thanks for your instruction and advice. We strongly agree with you. Although we provided the interacting evidence by Y2H, further experiments such as such as BiFC or Co-IP should be performed to confirm the interaction. We revised the Discussion to try to avoid of any inclination that the identified proteins are actually interacting with SCMV-6K2. Please check Line 268, 291-293, 344-350.

Point 5: Please check any grammatical errors. I noticed two such errors in Figure. 1 legend.

Response: The legend of Figureure. 1 was carefully revised and the manuscript was polished by the ‘MDPI Author Services’. Please check Line 71-75 for Figure. 1 legend.

Reviewer 2 Report

This is a well designed study with interesting results. However, before the paper can be published some parts of the paper must be improved.

The novelty and general significance of the paper should be clearly indicated (Introduction).

'Introduction' is not concluded by a clear hypothesis, which will be tested and subsequently discussed.

Which leaves/parts of the leaf were taken for measurements?

Which leaves (fully developed or young) were used for measurements? It is important to know whether host factors interacting with the 6K2 are age dependant and/or specific for the developmental stage or leaves chosen.

Plant growth conditions, especially light conditions (PPFD???), air humidity, are not described, please add.

What constituted a replicate? Were measurements based on one leaf or one group of leaves? It is unclear how plant samples were collected for analyses.

Unclear description for Figure. 1.

Leaves with typical mosaic symptoms should be added into manuscript.

The discussion needs to also include the limitations of this study and further experiments to answer important questions resulting from this study. Are there any implications for future research?

Author Response

Point 1: This is a well designed study with interesting results. However, before the paper can be published some parts of the paper must be improved. The novelty and general significance of the paper should be clearly indicated (Introduction).

Response: Thank you very much for your encouragement and advice. We described the novelty and significance of the manuscript in the Introduction section. Please check Line 56-57.

Point 2: 'Introduction' is not concluded by a clear hypothesis, which will be tested and subsequently discussed.

Response: Thank you very much. It has been accepted viral proteins extensively interact with host factors to establish systemic infection. However, it will be more rigorous and logical to give a hypothesis as the screening and identification of plant proteins interacting with potyviral 6K2. We revised the Introduction and Discussion section according to your advice. Please check Line 52-53, 338-339.

Point 3: Which leaves (fully developed or young) were used for measurements? It is important to know whether host factors interacting with the 6K2 are age dependent and/or specific for the developmental stage or leaves chosen.

Response: Many thanks for your advice. It is rigorous to consider the developmental stage of the sampled leaves used for cDNA library construction as the developmental stage of leaves imposes potential influence on the capacity of cDNA library, thereby affecting the screening results of library. The sugarcane cultivar ROC22 infected by SCMV-FZ1 was used in this study. To construct a cDNA library containing as many genes as possible, we sampled leaves showing typical mosaic symptom at seeding, tillering, fast-growing and sucrose accumlating stage, respectively. Then the sampled leaves were equalized to isolate RNA for cDNA library construction based on the DUALmembrane system. Thus the influence of leaves in different developmental stage on the results of library screening was decreased as possible as we can, although it is impossible that the library covers all the genes interacting with SCMV-6K2. As the construction, evaluation of the cDNA library was relative complicate and an independent work, we want to publish it in another paper. For gene cloning in the present study, the leaves of ROC22 showing typical mosaic symptom was used. Please check Line 347-350.

Point 4: Plant growth conditions, especially light conditions (PPFD???), air humidity, are not described, please add.

Response: Thank you very much. We described the plants culture condition in details. Please check Line 357-363.

Point 5: What constituted a replicate? Were measurements based on one leaf or one group of leaves? It is unclear how plant samples were collected for analyses.

Response: We sampled leaves with typical mosaic symptom for RNA isolation to clone gene. For verification of the interaction between SCMV-6K2 with the identified proteins individually, we performed triplicates in the Y2H assays. We revised the manuscript. Please check Line 431-432.

Point 6: Unclear description for Figure. 1.

Response: The legend of Figure. 1 was carefully revised and the manuscript was polished by the ‘MDPI Author Services’. Please check Line 71-75 for Figure. 1 legend.

Point 7: Leaves with typical mosaic symptoms should be added into manuscript.

Response: We provided the picture as Figure. S1 showing the typical mosaic symptom on the leaves of sugarcane cultivar ROC22. Please check the Supplementary File.

Point 8: The discussion needs to also include the limitations of this study and further experiments to answer important questions resulting from this study. Are there any implications for future research?

Response: Thank you very much. We emphasized the interaction evidence by Y2H is in vitro and made it clear that further experiments should be conducted to verify the interaction between the SCMV-6K2 with the twenty identified proteins in vivo at the end of the Discussion section. We also address the future work to illuminate function of the identified proteins in SCMV infection on sugarcane plants. Please check Line 344-351.